# Effects of antidiabetic agents on Alzheimer's disease biomarkers in experimentally induced hyperglycemic rat model by streptozocin

**Shatw Khalid Ali**[ID]*[◉], **Rojgar H. Ali**[ID][◉]

Department of Pharmacology and Toxicology, Hawler Medical University, Erbil, Iraq

[◉] These authors contributed equally to this work.
* shatwla@yahoo.co.uk

**Data Availability Statement:** All relevant data are within the manuscript and its Supporting Information files.

## Abstract

### Background

Alzheimer's disease is the most common cause of dementia in the elderly population. It is characterized by the accumulation of amyloid β and intraneuronal neurofibrillary tangles in the brain. Increasing evidence shows that the disturbance of insulin signalling in the brain may contribute to the pathophysiology of Alzheimer's disease. In type 1 diabetes, these disruptions are caused by hypoinsulinemia, but in type 2 diabetes, they are caused by insulin resistance and decreased insulin secretion. Multiple studies have shown that diabetes is connected with an increased risk of acquiring Alzheimer's disease. The aim of this study was to investigate the impact of anti-diabetic agents on Alzheimer's disease progression and the levels of Alzheimer's biomarkers in a hyperglycaemic rat model, which was induced by intraperitoneal injection of streptozocin to produce insulin-deficient diabetes.

### Method

Thirty-six male Wistar albino rats were allocated into six groups of six rats each. Group I was the negative control group. Intraperitoneal injections of streptozocin (42mg/kg) were used once for the five experimental groups. Group II served as the positive control group. The rats in Groups III, IV, V, and VI received metformin (300mg/kg), donepezil (10mg/kg), insulin glargine (3 unit/animal), and glibenclamide (10mg/kg), respectively, for 21 days.

### Results

Inducing hyperglycaemia in rats significantly increased the levels of serum glucose, haemoglobin A1c, total cholesterol, triglycerides, high-density lipoprotein, interleukin 6, tumour necrosis factor alpha, amyloid β 42, total plasma tau, and neurofilament light. A significant increase was also found in brain amyloid β 42, nitric oxide, acetylcholinesterase, malondialdehyde, β secretase, and phosphorylated microtubule-associated protein tau. The greatest statistically significant reductions in serum glucose, haemoglobin A1c, triglycerides, amyloid β 42, total plasma tau, brain amyloid β 42, acetylcholinesterase, and malondialdehyde were observed in rats treated with metformin. In contrast, rats treated with donepezil

**Funding:** The author(s) received no specific funding for this work.

**Competing interests:** The authors have declared that no competing interests exist.

demonstrated the greatest statistically significant reduction in serum tumour necrosis factor alpha, brain nitric oxide, and β secretase. The levels of neurofilament light and phosphorylated microtubule-associated protein tau in the brains of rats treated with insulin glargine were significantly lower than the other treatment groups. The total cholesterol and low-density lipoprotein levels in rats treated with glibenclamide exhibited the most statistically significant reductions of all the treatment groups.

## Conclusions

Metformin and donepezil, when administered at appropriate doses, were shown to successfully lower most plasma and brain biomarkers, including glucose, triglycerides, tumour necrosis factor alpha, amyloid β 42, nitric oxide, acetylcholinesterase, malondialdehyde, and β secretase in rats suffering from Diabetes Mellitus. As a result of this research, we suggest that metformin, either alone or in conjunction with donepezil, might be an excellent drug of choice for neuro-regeneration and risk reduction in Alzheimer's like disease.

## Introduction

Alzheimer's disease (AD) is the most common cause of dementia in the elderly population. It is characterized by neurodegeneration affecting both the cortex and the limbic system, as well as the accumulation of amyloid β and intraneuronal neurofibrillary tangles in the brain [1]. Despite various hypotheses, the specific pathologic pathways that cause neurodegeneration in AD are still unclear. Increasing evidence shows that the disturbance of insulin signalling in the brain may contribute to the pathophysiology of AD [2]. Several studies have discovered decreased insulin levels and insulin receptor expression in the brains of AD patients [3, 4], whereas other studies have highlighted insulin resistance [5], but all the evidence points to a breakdown of the insulin-signalling system. In addition to regulating food intake and energy balance, insulin and insulin receptors in the brain play a role in cognitive performance [6].

In type 1 diabetes mellitus (T1DM), these disruptions are caused by hypoinsulinemia, but in type 2 diabetes mellitus (T2DM), they are caused by insulin resistance and decreased insulin secretion. Multiple studies have shown that DM is connected to an increased risk of acquiring AD [7–9]. Although other studies have found no clear association between DM and cognitive impairment, dementia, or AD [10, 11], they emphasize that diabetes should be considered a potential risk factor for these conditions [12].

Disorders of the insulin-signalling pathway are emerging as common hallmarks of both AD and DM; nevertheless, the emphasis thus far has been on T2DM, with hyperinsulinemia and insulin resistance serving as the key insults [12]. Little information is known about the relationship between T1DM and AD, despite the fact that hypoinsulinemia causes a comparable impairment in insulin signalling. Furthermore, cognitive deficits, such as impaired learning and memory and difficulties with problem solving and mental flexibility, are more common in individuals with T1DM than in the normal population [13, 14], suggesting that the negative effect of cerebral hyperglycaemia or hypoinsulinemia exists in hyperglycaemic individuals.

As a result of these deficiencies, degradation of the cerebral cortex [15] and neuronal loss [16] are often identified at autopsy, and these findings are more evident in individuals with T1DM than in age-matched non-diabetic patients.

Recently, it was found that learning impairments are associated with increased tau phosphorylation, and higher amyloid β protein levels in the brain were identified in a T1DM

mouse model [17]. Similar alterations have also been seen in a spontaneous T1DM rat model [18].

To further investigate the roles of insulin deficiency and hyperglycaemia in the central nervous system dysfunction and pathology, we have investigated the impact of anti-diabetic agents on AD progression and AD biomarker levels in a hyperglycaemic rat model, which was induced by streptozocin (STZ) to produce insulin-deficient diabetes. Alterations in amyloid β 42, tau, and phosphorylated microtubule-associated protein tau (pMAPT/ptau), which are the key neuropathological markers of AD, were specifically examined in both the brain and serum. Furthermore, we investigated whether T1DM effects levels of interleukin 6 (IL6), tumour necrosis factor alpha (TNFa), neurofilament light (NFL), nitric oxide (NO), acetylcholinesterase (AChE), malondialdehyde (MDA), and β secretase.

## Method

Thirty-six male Wistar albino rats, weighing 200 to 300 g and aged 5 to 6 months, were evaluated in this study. One week prior to the start of the experiment, six male rats were housed per cage at 24˚C and 55% ± 5% humidity with a 12-hour light-dark cycle (light on at 8:00 am and light off at 8:00 pm). Unlimited access to water and ordinary rat pellet diets were provided. The animals were acquired via the Hawler Medical University, College of Pharmacy (Iraq-Erbil). This work was authorized by the ethics committee of the Hawler Medical University, College of Pharmacy, with permission number HMUPH-EC20211129-361. For the entire duration of the research project, we adhered to the "Guide for the Care and Use of Laboratory Animals," which was developed by the National Academy of Science and published by the National Institutes of Health.

## Materials

Rat TNFa, amyloid β peptide 42, AChE, NO, β secretase, MDA, (pMAPT/ptau), tau protein, NFL polypeptide, and rat IL6 enzyme-linked assay (ELISA) kits were purchased from Shanghai Sunred Biological Technological Co., Ltd (China). STZ was bought from Glentham Life Sciences Corsham, S13 0SW, United Kingdom. Glucophage (metformin; 500mg tab), Daonil (glibenclamide; 5mg tab), Lantus (insulin glargine; 100 units/ml pen), and Aricept (donepezil; 10mg tab) were purchased in a pharmacy and were manufactured by Merck Healthcare (Darmstadt, Germany), Sanofi (Paris, France), Sanofi (Paris, France), and Pfizer (New York, United States), respectively.

## Study design

In this study, the 36 male Wistar albino rats were allocated into six groups with a simple random sampling method (n = 6): Group I served as a negative control group, and Group II functioned as a positive control group.

On Day 1 of the experiment, all the rats were fasted for 12 hours prior to receiving STZ therapy and were given water as usual. A total of 21 mg of STZ was weighed into a 1.5-ml microcentrifuge tube, which was sealed with aluminium foil. One tube was used for each rat. The STZ was dissolved in 50 mM of sodium citrate buffer (pH 4.5) to a final concentration of 21 mg/ml immediately before injection. For groups II, III, IV, V, and VI, the STZ solution was injected intraperitoneally (i.p.) at 42 mg/kg (2.0 ml/kg) using a 1-ml syringe and a 23-G needle [19]. The negative control group received an equivalent amount of citrate buffer (pH 4.5) injected i.p. [20]. After injection, they were provided with normal food and maintained on 10% (w/v) sucrose water. On Day 2 of the experiment, the sucrose water was replaced with regular tap water [20].

On Day 3, all the rats were fasted for 6–8 hours in the early morning. The animals were anesthetized with 1.0 ml of xylazine (20 mg/ml), which was added to 10 ml of ketamine (100 mg/ml) and diluted by one-tenth with sterile saline [21]. Blood was collected through the medial canthus of the orbit using the end of a microhematocrit tube [22], and blood glucose concentration was measured using the Accu-Chek Instant blood glucose monitoring system to check for hyperglycaemia. More than 50% of the rats had blood glucose levels of more than 150 mg/dl (8.3 mmol/L), which was significantly higher than the control animals, suggesting that they were in the early stages of T1DM. On Day 10 of the experiment, the groups who failed the initial test for diabetes were tested again by repeating the above-mentioned procedure, and all of the rats had statistically higher blood sugar levels than the negative control group.

Most of the STZ-injected rats developed severe diabetes after 3 weeks, with blood glucose concentrations commonly ranging from 250 to 600 mg/dl (13.9–33.3 mmol/L) in most cases. According to previous studies, all of the DM complications should be apparent within 4–8 weeks [20]. Table 1 shows the variations over several days in rat blood glucose levels after STZ (42 mg/kg) injections.

Animals began treatment on the Day 61 of the experiment.

Group III: Received an oral daily dose of metformin, 300 mg/kg [23, 24].

Group IV: Received an oral daily dose of donepezil, 10 mg/kg [25].

Group V: Received a subcutaneous injection of insulin glargine, 3 units/rat [26].

Group VI: Received an oral daily dose of glibenclamide, 10 mg/kg [27, 28].

The oral medications were dissolved in normal saline, and the correct dosage was calculated. The medications were administered to the animals through oral gavage, except for insulin, which was injected subcutaneously.

After 21 days of drug administration, all animals were anesthetized with i.p. injection of xylazine (10 mg/kg) and ketamine (125 mg/kg) [29]. Blood was collected using the cardiac puncture method [22]. After centrifuging the blood samples, the separated serums were kept in tubes and put in the freezer at -23˚C for five days until the day the tests were measured. The animals were sacrificed via cervical dislocation at the end of the experiment. To confirm the animals' deaths, the heartbeats and pupillary reactions to light were measured. The brains were removed and the brain tissues were washed with ice-cold normal saline, mixed with phosphate buffer, and homogenized by hand by using mortar and pestle. The brain tissue extracts were separated by a cold centrifuge and placed in Eppendorf tubes, which were labelled with the group name and kept in a freezer at -80˚C for five days until the day the tests were measured.

## Biochemical assays

Rat TNFa, amyloid β peptide 42, AChE, NO, β secretase, MDA, pMAPT/ptau, tau protein, NFL, and IL6 were measured using ELISA kits specific for rats. Serum glucose, haemoglobin

**Table 1. High blood glucose levels in rats after exposure to STZ.** Hyperglycaemia was induced in rats using a single i.p. dose of STZ (42 mg/kg). Blood glucose levels were checked before and after the STZ injection.

| Experimental Day | Blood Glucose Mean ± SD |
| --- | --- |
| Day 0 | 79.38 mg/dl ± 3.20 |
| Day 3 | 178 mg/dl ± 16.30 |
| Day 10 | 234 mg/dl ± 13.70 |
| Day 30 | 367 mg/dl ± 27.19 |
| Day 60 | 537 mg/dl ± 43.51 |

A1c (HbA1c), triglycerides (TG), total cholesterol, high-density lipoprotein (HDL), and low-density lipoprotein (LDL) were measured using specific reagents for each of the parameters mentioned, using the Cobas-Roche analyser.

## Statistical analysis

All data were expressed as mean ± standard error mean (mean ± SEM). Differences in all the parameters between control and medication-treated rats were examined using statistical package for social science (SPSS) (version 25), and a one-way analysis of variance (ANOVA) was conducted to assess the results. A multiple comparison (Tukey) test was performed to compare the groups. A $P$ value < 0.05 was used to determine statistical significance. The datasets used or analysed for the present investigation are available from the corresponding author without restriction.

## Results

### The effects of different treatment groups on serum biomarkers

The results (Table 2) showed that the rats in the positive control group (pre-treated with STZ) had significantly higher blood glucose levels (761 ± 37.95 mg/dl) than the negative control group (132.16 ± 1.66 mg/dl) ($P$ < 0.05). A post hoc test was performed to discover the exact difference and significance level between the positive control group and the study groups. All forms of the treatments significantly reduced blood glucose levels compared to the positive control, except for donepezil ($P$ = 0.07) and insulin glargine ($P$ = 0.26).

The results indicated that the positive control group had significantly higher blood HbA1c levels (9.68 ± 0.8%) than the negative control group (5.51 ± 0.047%) ($P$ < 0.05). A post hoc test was performed to determine the exact difference and significance level between the groups (inter-group variation). The difference between all the groups after 21 days of treatment was not significant ($P$ > 0.05).

**Table 2. The effects of metformin, donepezil, insulin glargine, and glibenclamide on serum biomarkers: Glucose, HbA1c, lipid profile, IL6, TNFa, amyloid β 42, total plasma tau, and NFL.**

| Serum parameters | Negative control | STZ/positive control | STZ + metformin | STZ + donepezil | STZ + insulin glargine | STZ + glibenclamide | *P* value (ANOVA) |
|---|---|---|---|---|---|---|---|
| Glucose (mg/dl) | 132.16±1.66 | 761±37.95* | 326.83±17.06# | 676.33±9.02 | 695.66±22.74 | 450.16±16.61# | 0.0001 |
| Haemoglobin A1c (%) | 5.51±0.047 | 9.68±0.85* | 8.56±0.38 | 10.13±0.37 | 9.4±0.57 | 8.55±0.36 | 0.0001 |
| Total cholesterol (mg/dl) | 41.16±2.76 | 76.5±1.05* | 60±1.15# | 56.83±5.86# | 61.16±2.63# | 41.16±1.19# | 0.0001 |
| Triglycerides (mg/dl) | 60.16±3.41 | 128.33±26.69* | 61±10.48# | 65.33±1.38# | 68.16±3.19# | 74.16±1.88# | 0.002 |
| Low density lipoprotein (mg/dl) | 10.76±3.09 | 14.36±2.16 | 13.33±2.73 | 11.63±4.27 | 13.53±3.12 | 5.73±2.29 | 0.392 |
| High density lipoprotein (mg/dl) | 29.5±0.76 | 34.16 ± 2.44 | 34.83±0.54 | 34.16±2.74 | 36±1.84 | 24.5±1.11# | 0.001 |
| Interleukin 6 (pg/ml) | 32.28±3.31 | 81.6±0.75* | 92.1±1.53 | 106.91±11.99 | 117.4±8.86# | 95.98±2.38 | 0.0001 |
| Tumour necrosis factor a (ng/L) | 11.93±0.61 | 44.71±1.17* | 22.26±1.67# | 21.20±1.99# | 63.06±10.17# | 28.38±2.46 | 0.0001 |
| Amyloid β 42 (pg/ml) | 71.91±0.85 | 700.1±20.04* | 477.16±7.48# | 546.16±5.82# | 577.66±8.11# | 676±12.42 | 0.0001 |
| Total plasma tau (ng/L) | 22.93±0.51 | 140.83±8.04* | 90.16±1.47# | 115.06±1.67# | 93.9±2.05# | 106.9±7.17# | 0.0001 |
| Neurofilament light (ng/L) | 3.42±0.15 | 18.41±0.14* | 9.83±0.38# | 7.43±1.11# | 3.195±0.15# | 4.38±0.27# | 0.0001 |

Values are expressed as mean ± SEM

*$P$ < 0.05 when compared with the negative control group

#$P$ < 0.05 when compared with the positive control group

The results showed that a statistically significant relationship existed between the various treatment groups' serum total cholesterol levels. When a post hoc test was performed, it indicated that a significant reduction existed in all the treatment groups compared to the positive control group.

The findings also showed that the positive control group had significantly higher triglycerides levels ($128.33 \pm 26.69$ mg/dl) than the negative control group ($60.16 \pm 3.41$ mg/dl) ($P < 0.05$). A post hoc test was performed to determine the exact difference and significance level between the control and treatment groups, it indicated that a significant reduction existed in all the treatment groups compared to the positive control group ($P < 0.05$).

The results (Table 2) showed that no statistically significant association existed between the different treatment protocols and serum LDL levels. The mean serum LDL levels were similar in each group, ranging from 10–14 mg/dl. ANOVA was performed to compare the average serum LDL levels of all the groups ($P = 0.392$). A post hoc (Tukey) test was performed to identify the exact difference in HDL levels between the control and treatment groups. Except for glibenclamide, which differed substantially from all other treatment groups ($P < 0.05$), there was no significant difference across the groups.

Hyperglycaemia induction by STZ resulted in a significant increase of IL6 levels ($81.6 \pm 0.75$ pg/ml) ($P < 0.05$) compared to the negative control group ($32.28 \pm 3.3$ pg/ml). A post hoc (Tukey) test was performed, and the difference between all the groups was not significant ($P > 0.05$), except for positive control and insulin glargine ($P < 0.05$), which increased IL6 levels significantly more than the positive control and all the treatment groups.

Plasma TNFa concentration in the STZ-pre-treated rats was significantly higher ($44.71 \pm 1.17$ ng/L) than that in the negative control group ($11.93 \pm 0.61$ ng/L) ($P < 0.05$). The rats treated with insulin had significantly highest plasma TNFa levels compared to the positive control group and among the other treated groups, the rats treated with metformin and donepezil had significantly lowest TNFa levels compared to the positive control group ($P < 0.05$).

The i.p. injection of STZ in the positive control group significantly raised the serum level of amyloid β 1–42 ($700.1 \pm 20.04$ pg/ml) compared to the negative control group ($71.91 \pm 0.85$ pg/ml) ($P < 0.05$). A post hoc test was performed, and there was a significant increase in all the treatment groups compared to the positive control group ($P < 0.05$), except for glibenclamide ($P = 0.62$).

The rats pre-treated with STZ, the serum total plasma tau protein was significantly higher ($140.83 \pm 8.04$ ng/L) than that in the control group ($22.93 \pm 0.51$ ng/L) ($P < 0.05$). A post hoc test was performed, a significant reduction existed in all the treatment groups compared to the positive control group ($P < 0.05$).

In the hyperglycaemia-induced rats, serum NFL levels were higher ($18.41 \pm 0.14$ ng/L) compared to the negative control group ($3.42 \pm 0.15$ ng/L) ($P < 0.05$). A post hoc test indicated that a significant reduction existed in all the treatment groups compared to the positive control group ($P < 0.05$).

## The effects of different treatment groups on brain biomarkers

The results (Table 3) showed that the control group (pre-treated with STZ) had significantly higher brain amyloid β 42 levels ($778 \pm 15.85$ pg/ml) than the negative control group ($86 \pm 1.96$ pg/ml) ($P < 0.05$). A post hoc test was performed to determine the exact difference and significance level between the positive control group and the treatment groups, a significant reduction existed in all the treatment groups compared to the positive control group ($P < 0.05$).

**Table 3. The effects of metformin, donepezil, insulin glargine, and glibenclamide on brain biomarkers: Amyloid β 42, nitric oxide, acetylcholinesterase, malondialdehyde, β secretase, and pMAPT.**

| Brain parameters | Negative control | STZ/positive control | STZ + metformin | STZ + donepezil | STZ + insulin glargine | STZ + glibenclamide | P value (ANOVA) |
|---|---|---|---|---|---|---|---|
| Amyloid β 42 (pg/ml) | 86±1.96 | 778±15.85* | 443.16±9.54# | 531.76±9.81# | 546.66±8.78# | 703.73±4.16# | 0.0001 |
| Nitric oxide (umol/L) | 7.43±0.19 | 29.86±0.24* | 9.62±0.14# | 8.98±0.18# | 26.66±1.09# | 24.4±1.18# | 0.0001 |
| Acetylcholinesterase (ng/ml) | 41.48±0.56 | 84.90±1.36* | 46.25±0.77# | 53.81±0.88# | 53.63±0.93# | 72.3±0.51# | 0.0001 |
| Malondialdehyde (nmol/ml) | 0.35±0.041 | 4.32±0.16* | 0.97±0.01# | 3.15±0.24# | 2.65±0.13# | 3.55±0.19# | 0.0001 |
| β secretase (pg/ml) | 116.36±0.94 | 862.03±8.38* | 245.58±5.67# | 187.05±1.94# | 317.98±1.06# | 745.53±3.60# | 0.0001 |
| pMAPT (ng/L) | 16.38±0.68 | 44.55±0.89* | 24.26±0.64# | 34.24±1.23# | 18.53±0.38# | 20.81±1.14# | 0.0001 |

Values are expressed as mean ± SEM

*$P < 0.05$ when compared with the negative control group

#$P < 0.05$ when compared with the positive control group

The brain nitric oxide level in STZ pre-treated rats was significantly higher (29.86 ± 0.24 μmol/L) than in the control group (7.43 ± 0.19 μmol/L) ($P < 0.05$). The brain nitric oxide level was significantly reduced in all the treatment groups ($P < 0.05$).

The level of AChE in hyperglycemia-induced rats was significantly higher than in the negative control group. A post hoc test was performed, and the AChE level was significantly reduced in all the treatment groups ($P < 0.05$).

The level of MDA was significantly higher in hyperglycaemia-induced rats than in the negative control group. According to the post hoc analysis, the brain MDA level was significantly reduced in all the treatment groups ($P < 0.05$).

The level of β secretase was significantly higher in hyperglycemia-induced rats than in the negative control group. A post hoc test was performed, and it revealed that the brain β secretase level was significantly reduced in all the treatment groups ($P < 0.05$).

The level of pMAPT in the brains of STZ pre-treated rats was significantly higher (44.55 ± 0.89 ng/L) than in the control group (16.38 ± 0.68 ng/L) ($P < 0.05$). Brain pMAPT levels were significantly reduced in all treatment groups ($P < 0.05$).

## Discussion

Previously, AD and DM were seen as two distinct diseases. In contrast, a number of clinical and pre-clinical investigations have shown that AD and DM have comparable pathogenic pathways [30]. However, DM cannot be assumed to be sufficient to cause AD; rather, it may play a role as a cofactor in the progression of the illness due to abnormalities in insulin signaling that are accompanied by considerable elevation of amyloid β aggregation, tau hyperphosphorylation, inflammation, oxidative stress, and mitochondrial dysfunction [31]. Due to the same pathogenesis, it has been proposed that anti-diabetic medications may have therapeutic promise for the treatment of AD. Clinical investigations are now investigating these theories. It was previously known that some anti-diabetic drugs were helpful against a number of the hallmark AD pathologies, and it was also recognized that these treatments increased neurogenesis. Although the outcomes of these medications are promising, a thorough understanding of the shared pathomechanisms between DM and AD, the central and peripheral molecular actions of medications, and the impact of demographic changes and genetic mutations on AD development is urgently required for diagnostic and therapeutic purposes [31]. Additionally,

an increase in AD knowledge, diagnosis, and possible treatments, as well as population-based research, are crucial for preventing dementia caused by diabetes in high-risk populations.

In mammals, STZ is highly toxic to the insulin-producing beta cells of the pancreas. It has the potential to selectively destroy the cells responsible for generating and releasing insulin, and it is used in medical research to establish an animal model for T1DM when administered in high dosages. STZ administration by a method such as intracerebroventricular or i.p. injection results in decreased cognition and increased cerebral aggregated amyloid β fragments, total tau protein, and amyloid β deposits [32]. Tables 2 and 3 show that i.p. administration of STZ (42 mg/kg) in rats caused significant physiological alterations that can be compared to AD.

Blood glucose level is a sensitive indication of DM, and rats with a baseline fasting blood glucose level <135 mg/dL were regarded as normal [33]. After STZ administration, blood glucose levels were considerably higher in this study. The damage to pancreatic cells that STZ caused led to hypoinsulinemia and hyperglycaemia. The fact that the elevated level persisted after 21 days of therapy indicates that the harm was irreversible.

The glycosylated haemoglobin (HbA1c) test reflects blood glucose concentrations during the previous 8–12 weeks and is often used as an indicator of average blood glucose concentration. When the HbA1c test is less than 5.7%, it is considered normal [34]. In this investigation, the HbA1c level was considerably higher than that following STZ administration. An increase in HbA1c was caused by establishing a sugar-haemoglobin connection, which indicates the presence of excessive sugar in circulation, a sign of diabetic complications [35]. This discovery is consistent with the findings of previous research in the field [34].

Lipids, as the fundamental building blocks of cell membranes, play a significant role in human health and brain function. The brain contains a high concentration of lipids, and the dysregulation of lipid homeostasis is related to aging and plays a significant role in the aetiology of AD [36]. A lipid profile is a blood test that detects problems in lipid levels and mainly measures HDL, LDL, triglycerides, and total cholesterol [37, 38]. According to the findings of this research, glibenclamide had a significant effect on lowering HDL levels in all treatment groups when compared to the positive control group and LDL levels were not significantly different amongst the groups. All the treatment groups were able to significantly lower triglycerides levels, with metformin being the most successful. This finding is in accordance with the results of past research studies [39]. All the treatments were effective in lowering total cholesterol compared to the positive control group, particularly glibenclamide, and this finding is also consistent with previous research [40].

IL6 is a pleiotropic cytokine that plays a critical role in host defence, owing to its extensive spectrum of immunological and hematologic activities and its strong capacity to elicit the acute phase response [41]. In a recent study, both mild and moderately severe AD patients had significantly higher IL6 secretion levels than healthy individuals of the same age. These elevated levels of peripheral IL6 secretion might have been responsible for the acute-phase proteins in their serum [42]. In the current study, after induction of DM, none of the treatment groups reduced the level of IL6 below that of the positive control group.

TNFa is a gliotransmitter that regulates synaptic activity in brain networks, and it has recently been shown to have an important role in the breakdown of synaptic memory pathways caused by amyloid β and amyloid-β oligomers [43]. The current research showed that DM significantly increased TNFa levels and among all the medications, metformin and donepezil could significantly reduce TNFa levels when compared to the positive control group. This conclusion is in line with earlier study findings [44, 45].

Tau is a protein that controls the formation of microtubules and their structural integrity under physiological conditions [46]. Additionally, amyloid β is well known for forming

amyloid plaque on nerve cells in the brains of AD patients [47]. According to studies, tau phosphorylation and amyloid β buildup both have a role in the development and pathophysiology of AD [47]. There is also a relationship between tau and DM, since both insulin and insulin growth factor 1 are involved in tau phosphorylation, which is linked to the formation of neurofibrillary tangles and synaptic loss. When insulin communication in the brain is disrupted, tau phosphorylation begins, resulting in reduced Akt kinase activity (also known as protein kinase B) and increased glycogen synthase kinase 3 beta (GSK-3) activity [48]. GSK-3 regulates glucose levels in the blood by contributing to glycogen production [49]. Increased levels of total tau and phosphorylated tau were identified in the brains of both T1DM and T2DM patients compared to healthy controls [50, 51]. In addition, T1DM patients have a greater concentration of amyloid β peptide 42 in their brains than healthy controls [51]. Finally, advanced glycation end (AGE) products are formed as a result of dysregulated glucose metabolism in DM. Increased AGE levels in the brain have been observed to enhance amyloid β 42 aggregation by preventing its clearance. The density of AGE receptors is enhanced in AD and is engaged in amyloid β related inflammatory processes [52].

Several clinical trials have shown that diabetic people who use metformin for a long time have greater cognitive function than those who take other anti-diabetic medicines. Metformin has been shown to reduce tau phosphorylation and amyloid β formation by inhibiting AChE activity and therefore raising acetylcholine content in the brain, which is important for learning and memory [53]. Overall, it seems that the commonly used AD biomarkers (tau protein hyperphosphorylation and amyloid β accumulation) are linked to memory impairment in people with DM, suggesting that these biomarkers may be useful in DM patients.

In this study, the level of amyloid β in serum and brain tissue was significantly increased after the induction of DM. Among all the treatment groups, particularly metformin, had significantly reduced amyloid β levels relative to the positive control group. This result is in line with previous studies that suggest that metformin may be an ideal choice for neuro-regeneration and risk reduction of AD [54]. The results of this study on plasma tau show that after the induction of DM, a significant rise in plasma tau levels occurred. All the treatment groups were associated with significantly lower tau levels than the positive control group, with metformin and insulin glargine being the most effective. This finding is similar to that reported in a previous study on metformin [55].

It has recently been proposed that the NFL chain is a neuron-specific structural protein that can be measured in cerebrospinal fluid and plasma [56, 57] to detect axonal injury and neurodegeneration in a wide variety of neurological disorders, including AD [58]. Higher NFL levels have been associated with increased mortality in AD and other neurodegenerative disorders [59]. The current results indicate that all the treatment groups were able to significantly reduce NFL levels, with insulin glargine and glibenclamide being the most effective in returning the levels to normal.

NO is an enzymatic product of NO synthase that has major physiological activities. An accumulating body of data shows that NO pathways are linked to a range of neurological illnesses, including AD and other neurodegenerative dementias. Aging with a vascular risk factor reduces cerebral blood flow, resulting in microvasculopathy with reduced NO release and localized metabolic dysfunction [60]. This study found that across all the treatment groups, only metformin and donepezil were able to significantly reduce NO levels to normal, which is consistent with results from other studies [61, 62].

AChE is a cholinergic enzyme present in postsynaptic neuromuscular junctions, especially in muscles and nerves. Several studies have revealed that AChE activity and its molecular forms differ in AD tissues. Many of these studies examined whether differences in AChE isoform distribution can be used as a biochemical Alzheimer's diagnosis [63, 64]. In this study,

the level of AChE in the brain increased significantly after the induction of DM, and all treatments significantly lowered AChE levels compared to the positive control group, with metformin, donepezil, and insulin glargine being the most effective at returning the levels to near-normal, which is consistent with results from other studies [53, 65, 66].

MDA is a regularly used biomarker for lipid peroxidation and oxidative stress. Lipid peroxidation is a series of reactions that produce active molecules that cause cellular damage [67], and oxidative stress has been related to several diseases, including AD [68]. After induction of DM, MDA levels increased significantly, and only metformin returned the levels to near-normal. This result is in line with earlier research findings [69, 70].

β secretase (BACE1, or APP-cleaving enzyme 1) is an aspartic proteinase involved in cell differentiation, immunoregulation, and cell death. BACE1 is the major β secretase in neurons that produces amyloid-β peptides [71], and growing evidence links BACE1 activity changes to diseases like AD [72]. Following DM induction, the level of β secretase increased significantly, and among all the treatment groups, donepezil and metformin were the most successful in decreasing the β secretase level. This discovery also aligns with prior research findings [53, 73].

Tau proteins are crucial for microtubule stability in the body. These proteins are abundant in nerve cells but far less so in oligodendrocytes and astrocytes [74]. Pathologies of the nervous system, such as AD, may occur when tau proteins become deficient and fail to appropriately maintain microtubules [75]. The level of pMAPT significantly increased following the induction of DM, and insulin glargine was the only therapy that could return it to a normal level. This finding is in keeping with earlier results [76], although the other treatments only slightly reduced the pMAPT level compared to the positive control group.

## Conclusion

Metformin and donepezil, when administered at 300mg/kg and 10mg/kg, respectively, were shown lower most plasma and brain biomarkers, including glucose, triglycerides, tumour necrosis factor a, amyloid β 42, nitric oxide, acetylcholinesterase, malondialdehyde, and β secretase in rats suffering from diabetes mellitus. As a result of this research, we suggest that metformin, either alone or in conjunction with donepezil, might be an excellent drug of choice for neuro-regeneration and risk reduction in Alzheimer's like disease.

## Supporting information

**S1 Data.**
(DOC)

## Acknowledgments

The authors express their gratitude to Muhaeman Yusif, Rafel Abdulrazak, and Govand Tawfeeq for their efforts in providing the required facilities for the study to be carried out.

## Author Contributions

**Conceptualization:** Shatw Khalid Ali.

**Data curation:** Shatw Khalid Ali.

**Formal analysis:** Shatw Khalid Ali.

**Funding acquisition:** Shatw Khalid Ali.

**Investigation:** Shatw Khalid Ali.

**Methodology:** Shatw Khalid Ali, Rojgar H. Ali.

**Project administration:** Shatw Khalid Ali.

**Resources:** Shatw Khalid Ali.

**Software:** Shatw Khalid Ali.

**Supervision:** Rojgar H. Ali.

**Validation:** Shatw Khalid Ali.

**Visualization:** Shatw Khalid Ali.

**Writing – original draft:** Shatw Khalid Ali.

**Writing – review & editing:** Shatw Khalid Ali.

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
