## [Decision Letter · Decision Letter 0]

23 Mar 2022

PONE-D-22-05274Effects of Antidiabetic Agents on Alzheimer’s Disease Biomarkers in Rats with Experimentally Induced Diabetes: Randomized Experimental DesignPLOS ONE

Dear Dr. Ali,

Thank you for submitting your manuscript to PLOS ONE. After careful consideration by 2 Reviewers and an Academic Editor, all of the critiques of both Reviewers, especially the misnomer of the 'AD model', must be addressed in detail in a revision to determine publication status. If you are prepared to undertake the work required, I would be pleased to reconsider my decision, but revision of the original submission without directly addressing the critiques of the Reviewers does not guarantee acceptance for publication in PLOS ONE. If the authors do not feel that the queries can be addressed, please consider submitting to another publication medium. A revised submission will be sent out for re-review. The authors are urged to have the manuscript given a hard copyedit for syntax and grammar as this is requisite for publication consideration.

**Comments to the Author**

1. Is the manuscript technically sound, and do the data support the conclusions?

Reviewer #1: Partly

Reviewer #2: Partly

2. Has the statistical analysis been performed appropriately and rigorously? 

Reviewer #1: No

Reviewer #2: I Don't Know

3. Have the authors made all data underlying the findings in their manuscript fully available?

Reviewer #1: No

Reviewer #2: Yes

4. Is the manuscript presented in an intelligible fashion and written in standard English?

Reviewer #1: No

Reviewer #2: No

5. Review Comments to the Author

Reviewer #1: In this study, authors compared the effectiveness of multiple antidiabetic drugs in a model of experimental diabetes mellitus induced by intraperitoneal (ip) injection of streptozocin (STZ). A myriad of serum markers were then assessed in controls and type 2 diabetes animals.

1) The first issue with the study is that authors state in the Abstract that ip STZ injection induces Alzheimer's disease model. This is not true, because the sporadic AD-type model is induced by intracerebroventricular STZ injection. Authors should specify that with ip injection of STZ they induced diabetes-like changes and not AD.

2) Why do authors use abbreviations in brackets multiple times in the text? And why should "vascular dementia" and "cerebrovascular disease" be abbreviated if it is used only once in the whole manuscript?

3) Method: it is insufficient to state the weight of the animals. What was the age of animals? What conditions were they kept in (temperature, lighting, humidity)? How many rats per cage? Was there an acclimatization period? All of this information is missing. Authors state that blood glucose was measured "in the eyes" (line 110). I suppose this means that it was a retro-orbital blood collection? Was anaesthesia used for this? Procedure description is incomplete. What do authors mean in lines 110-111 by "and Accu Check Glucometer and left for 2 months". So animals were injected ip with STZ and left for 2 months, and only after that treated with various compounds for 21 consecutive days? It is unclear. A picture with experimental design would greatly aid to understanding the timeline of the procedures done by authors.

4) Results: abbreviation HSD is not defined. The use of the same symbol to depict differences between different groups seems misleading. Consider using * for comparison with negative control and # for comparison with positive control. Line 136 states that mean +/- S.E.M. was used, but in line 171 authors mention S.D. Which was it then, SEM or SD? In lines 178-179 authors state that HDL levels significantly differed in glibenclamide-treated STZ rats compared to all treatment groups. This is not shown in Table 1. Why would authors report ANOVA results for LDL levels but not for other markers? ANOVA tells us if the values between groups differ, and then by using a post-hoc test we show the values of which specific groups are significantly different. ANOVA is not really reported in this manuscript.

5) Discussion is too large. Description of each marker (and the rationale for assessing their levels) should be stated in the Introduction! In lines 261-264 authors describe the changes seen after peripheral injection of STZ, while citing an article on the effects of centrally injected STZ (Singh et al., 2015).

Overall, manuscript looks like it was not read thoroughly before submitting, as indicated by discrepancies in the use of abbreviation and grammar. Moreover, authors state that they induced AD, which they did not.

Reviewer #2: In “Effects of Antidiabetic Agents on Alzheimer’s Disease Biomarkers in Rats with Experimentally Induced Diabetes: Randomized Experimental Design”, the authors present a study utilizing a hyperglycemic rat model that also exhibits elevations in several biomarkers for Alzheimer’s Disease (AD). After 3 weeks of administering various diabetes and AD medications to different groups of rats, several metabolic and cognitive parameters are quantified in the serum and brain extracts. The findings of the study are relevant and potentially useful to the scientific and medical community. However, many concerning issues with the study first need to be addressed.

The authors perform intraperitoneal (i.p.) injections of STZ in 5 to 6 month old male rats to generate models of hyperglycemia and AD. While this procedure is perfectly valid for generating hyperglycemia, the legitimacy of this method for generating AD is questionable. The references the authors cite for inducing AD by this procedure do not support their methodology. For example, in reference 12, Zhang et al. induced learning and memory impairments by injecting STZ (i.p.) into neonatal mice. In order to induce such effects in adult mice, Zhang et al. administered intracerebroventricular STZ injections. Furthermore, although the current authors do show several biomarkers associated with AD (e.g. amyloid beta) to be upregulated in their experimental rat model, no cognitive or behavioral tests are ever performed. I suggest that the authors present this model as a hyperglycemic model rather than a model of AD, unless data can be added to the manuscript that verifies learning and memory deficits.

After induction of the rat model, the authors divided the rodents into various treatment groups and administered different therapeutic agents to each group for 21 days. Most of the utilized agents were anti-diabetic agents with one exception – donepezil – a typical AD agent. Two control groups were included (no STZ and STZ without further treatment). The authors then sacrificed the mice and quantified the amounts of various biomarkers in the serum and brain extracts using ELISAs. While the data presented are potentially useful to the scientific and medical community, several issues were identified that should be addressed.

• First, it appears as though the authors did not measure blood glucose in all of the rats before starting the various treatments. While unlikely, it is possible that the STZ did not effectively induce hyperglycemia in all rodents, and this could potentially skew the apparent results of any treatment that those rats received. If the authors did indeed verify hyperglycemia in all of the rats before beginning treatment, those data should be provided in the manuscript.

• Second, while this is a minor issue, most of the diabetes agents used in this study are type 2 diabetes agents (e.g. metformin and glibenclamide). It is unclear how relevant these agents are in the context of an STZ model, since STZ induces a state much more similar to type 1 diabetes. The authors do test one insulin analogue (insulin glargine) which is a long-acting analogue that is used in the context of type 1 diabetes. However, it would not be used alone in the setting of type 1 diabetes. It would typically be accompanied by mealtime boluses of regular or rapid acting insulin.

• The results from the ELISAs are all presented in two tables, which are well organized and clear. However, the description of the results in the text is overly repetitive. Much of this text could be condensed. Also, I am not sure if comparing each treatment to every other treatment for each biomarker in the statistical analysis adds value to the study. It is somewhat confusing. In my opinion, simply comparing the STZ control to the no STZ control and then each treatment group to the STZ control would be sufficient. That is how the data are presented in the tables.

• It would be nice for the authors to comment on the potential of the anesthetizing agent to affect biomarkers, especially the brain biomarkers.

Lastly, the authors continuously refer to the model as an AD model, but it is not clear that the model or the diabetes treatments would be relevant to AD in the context of normoglycemia. As mentioned above, it would be best to present this model as a model of hyperglycemia with upregulation of several AD associated biomarkers. The findings may be most relevant to diabetes and diabetes linked AD.

In addition to the scientific or presentation concerns mentioned above, the authors also need to heavily revise the manuscript for language and grammatical errors. There are too many errors for this reviewer to correct at this stage. However, a few key points are mentioned below:

• Many words are inappropriately capitalized (e.g. Glucose, Hemoglobin, Total Cholesterol, etc…).

• Once the abbreviation for a word is given in parenthesis, the parenthesis does not need to be used throughout the manuscript. For example, the authors routinely refer to AD as (AD).

• Lines 64 and 65: The statement “(AD) and vascular dementia have both been demonstrated to enhance the risk of cognitive impairment and dementia in people with type 2 (DM) (8,9).” is incorrect. After reviewing the sources cited (8 and 9), I suspect the authors meant “T2DM has been shown to enhance the risk of cognitive impairment, AD, and vascular dementia”.

• Line 114: Typo: “oral daily doses of Insulin”. The authors later state that insulin was given subcutaneously, which is the correct way to administer insulin glargine.

• Line 263: STZ does not induce insulin resistance.

Despite the many concerns raised in the above review, I do think that the findings have scientific merit and can be useful to the scientific community if presented in the right context.

6. PLOS authors have the option to publish the peer review history of their article (what does this mean?). If published, this will include your full peer review and any attached files.

**Do you want your identity to be public for this peer review?** For information about this choice, including consent withdrawal, please see our Privacy Policy.

Reviewer #1: No

Reviewer #2: No

We look forward to receiving your revised manuscript.

Kind regards,

Stephen D. Ginsberg, Ph.D.

Section Editor

PLOS ONE

---

## [Author Response · Author response to Decision Letter 0]

28 Apr 2022

Many thanks for your helpful comments, almost all the mistakes are fixed and almost all the manuscript has been revised upon your request. We hope that now the manuscript is in the right context. 

Also, our comments can be found in the (response to the reviewer) file.

---

## [Decision Letter · Decision Letter 1]

16 May 2022

PONE-D-22-05274R1Effects of Antidiabetic Agents on Alzheimer’s Disease Biomarkers in Experimentally Induced Hyperglycemic Rat Model by Streptozocin: Randomized Experimental Design

PLOS ONE

Dear Dr. Ali,

Thank you for resubmitting your work to PLOS ONE. Please make the corrections posed by Reviewer #2 so I can render a decision on this manuscript.

Please submit your revised manuscript by Jun 30 2022 11:59PM. If you will need more time than this to complete your revisions, please reply to this message or contact the journal office at plosone@plos.org. Please include the following items when submitting your revised manuscript:A rebuttal letter that responds to each point raised by the academic editor and reviewer(s). You should upload this letter as a separate file labeled 'Response to Reviewers'.A marked-up copy of your manuscript that highlights changes made to the original version. You should upload this as a separate file labeled 'Revised Manuscript with Track Changes'.An unmarked version of your revised paper without tracked changes. You should upload this as a separate file labeled 'Manuscript'.

We look forward to receiving your revised manuscript.

Kind regards,

Stephen D. Ginsberg, Ph.D.

Section Editor

PLOS ONE

Journal Requirements:

**Comments to the Author**

1. If the authors have adequately addressed your comments raised in a previous round of review and you feel that this manuscript is now acceptable for publication, you may indicate that here to bypass the “Comments to the Author” section, enter your conflict of interest statement in the “Confidential to Editor” section, and submit your "Accept" recommendation.

Reviewer #1: All comments have been addressed

Reviewer #2: (No Response)

2. Is the manuscript technically sound, and do the data support the conclusions?

Reviewer #1: Partly

Reviewer #2: Partly

3. Has the statistical analysis been performed appropriately and rigorously? 

Reviewer #1: Yes

Reviewer #2: I Don't Know

4. Have the authors made all data underlying the findings in their manuscript fully available?

Reviewer #1: Yes

Reviewer #2: Yes

5. Is the manuscript presented in an intelligible fashion and written in standard English?

Reviewer #1: Yes

Reviewer #2: Yes

6. Review Comments to the Author

Reviewer #1: (No Response)

Reviewer #2: In the latest revision of the manuscript entitled “Effects of Antidiabetic Agents on Alzheimer’s Disease Biomarkers in Experimentally Induced Hyperglycemic Rat Model by Streptozocin” the authors did address many of the concerns raised in the original review. In particular, the authors no longer refer to the model as an AD model, but rather as a model of DM, which was the major issue raised by both reviewers. The authors also added information about the blood glucose levels of the rats prior to treatment induction, as requested by Reviewer 2. They also attempted to simplify and clarify some of the results section by mostly comparing the treatment groups to the positive control group, as suggested. The English is also very much improved in the revised version. While the manuscript has indeed improved, it is my opinion that further improvements can be made. I have added further suggestions, line by line, below.

Line 5: “Randomized Experimental Design” likely not needed in the title.

Line 21: “… resulting in insulin resistance”. Insulin resistance is only present in T2DM, so this should be removed from the general statement about diabetes. The next line differentiates between T1DM and T2DM.

Line 63: “.. resulting in insulin resistance.” Not all diabetes has insulin resistance – just T2DM.

Line 71: Replace “diabetes” with “DM”.

Line 77: Did you mean to use the word “hypertensive”, or did you mean “hyperglycemic”?

Line 105: Change to “rat IL6 enzyme-linked assay (ELISA) kits” instead of “….. rat kits..”

Line 106: The parentheses around the company name is not needed.

Line 121: Consider rephrasing to make it clear that only groups II, III, IV, V, and VI received the STZ. Then the last sentence of this paragraph (Line 124 – 125) could be deleted.

Line 138: This sentence is confusing. Do you mean that according to previous studies, all of the complications should be apparent within 4 – 8 weeks? Or do you mean that all of the diabetes complications in this study were apparent in 4 – 8 weeks?

Table 1: Consider using averages and standard deviations rather than ranges.

Line 144: Consider replacing “were treated” with “began treatment” since the treatment was continuous.

Lines 149 – 151: The grammar needs correction. Suggestion: “The oral medications were dissolved in normal saline, and the correct dosage was calculated. The medications were administered to the animals through oral gavage, except for insulin, which was injected subcutaneously.”

Line 154: How were the serum tubes stored? At what temperature and for how long?

Line 158: “… homogenized by hand.” Was a mortar and pestle used? Or a tissue grinder? Can you specify? Also consider using the term “brain extracts” instead of “fluid of brain tissues”.

Line 160: Consider replacing the phrase “for a few days” with something more formal like “three days” or “approximately 3 days”.

Line 176: Consider changing “serum blood level” to “serum biomarkers”. Also, the colon is not needed in the section titles.

Lines 180 – 182: The way this is phrased is confusing. Are you saying that all forms of treatment significantly reduced blood glucose levels compared to the positive control, except for donepezil and insulin glargine?

Line 183: There is no need to keep repeating “(pre-treated with STZ)”, as that should be clear already when you refer to the positive control.

Lines 188 – 190: Can you specify what kind of difference (e.g. reduction)?

A similar issue exists in lines 191 – 195.

Lines 196 – 206: Since there are no differences in LDL or HDL, can these paragraphs be combined/condensed?

Line 204 – 205: If the ANOVA showed no significant difference, the analysis should stop there. How can one continue with the post hoc?

Line 207 – 211: Same issue as above with the post hoc following a non-significant ANOVA. Please clarify.

Line 214: The wording is confusing. I can see in the table that the insulin group had higher TNFa than the positive control, but this is not clear in the text.

Line 223: Again, the wording is confusing. Are you saying that all treatment groups exhibited significantly lower levels of tau in the serum compared to the positive control?

Line 226: Similar issue as that raised for line 223.

Line 236 and 237: What does it mean when you say “the difference between all the groups was significant”? Are you saying that all of the groups were significantly different than the positive control? That’s how the data are presented in the table. Also, can you specify in the text whether they were higher or lower? This type of issue occurs several times in the results section. Please clarify throughout the manuscript.

Table 3: The column titles are not consistent with those in Table 2. In Table 2, the column titles do not contain “STZ + ..” for the treatment groups. This is a minor issue.

Line 261: The wording here makes it sound like STZ is naturally occurring in mammals, which is not the case.

Line 272: Consider rephrasing. As it currently reads, it suggests that hyperglycemia is a cause of increased glucose levels, and those are the same things.

Line 285 – 287: Consider revising. As mentioned earlier, if the ANOVA was not significant, no further post hoc should be performed.

Lines 304 – 307: This is not clear. Consider rephrasing.

Line 312: The text says “all the treatment groups” and only later mentions that glibenclamide was an exception. Consider rephrasing.

General Comments:

The structure of the article and the way the results are presented is somewhat confusing or hard to follow. It is not clear why these particular medications were chosen or why these exact biomarkers were chosen. It is hard to keep track of what affects what as you are reading through the manuscript. The tables can be a useful reference for anyone interested in how particular medications have been shown to affect particular biomarkers, so I think the results can be useful.

A good place to bring things together and discuss the overarching relevance of the results would be the discussion section. However, I feel this is currently lacking, as the discussion section largely just restates the results.

Furthermore, while I believe that the statistics are sound for the most part, there are several instances in the results section where the data are presented in a way that does not fit with the described statistical methods. These instances are mentioned above in the “line by line” section.

7. PLOS authors have the option to publish the peer review history of their article (what does this mean?). If published, this will include your full peer review and any attached files.

**Do you want your identity to be public for this peer review?** For information about this choice, including consent withdrawal, please see our Privacy Policy.

Reviewer #1: No

Reviewer #2: No

---

## [Author Response · Author response to Decision Letter 1]

11 Jun 2022

I'm ready to edit and correct my manuscript if the manuscript returned back for request of correction or addition of an information.

---

## [Editor Report · Decision Letter 2]

24 Jun 2022

Effects of Antidiabetic Agents on Alzheimer’s Disease Biomarkers in Experimentally Induced Hyperglycemic Rat Model by Streptozocin

PONE-D-22-05274R2

Dear Dr. Ali,

We’re pleased to inform you that your manuscript has been judged scientifically suitable for publication and will be formally accepted for publication once it meets all outstanding technical requirements.

Kind regards,

Stephen D. Ginsberg, Ph.D.

Section Editor

PLOS ONE

---

## [Editor Report · Acceptance letter]

29 Jun 2022

PONE-D-22-05274R2 

Effects of Antidiabetic Agents on Alzheimer’s Disease Biomarkers in Experimentally Induced Hyperglycemic Rat Model by Streptozocin 

Dear Dr. Ali:

I'm pleased to inform you that your manuscript has been deemed suitable for publication in PLOS ONE. Congratulations! Your manuscript is now with our production department. 

Kind regards, 

on behalf of

Dr. Stephen D. Ginsberg 

Section Editor

PLOS ONE